# Long-LRM: Long-sequence Large Reconstruction Model for Wide-coverage Gaussian Splats

## Abstract

We propose Long-LRM, a generalizable 3D Gaussian reconstruction model that is capable of reconstructing a large scene from a long sequence of input images. Specifically, our model can process 32 source images at $960 \times 540$ resolution within only 1.3 seconds on a single A100 80G GPU. Our architecture features a mixture of the recent Mamba2 blocks and the classical transformer blocks which allowed many more tokens to be processed than prior work, enhanced by efficient token merging and Gaussian pruning steps that balance between quality and efficiency. Unlike previous generalizable 3D GS models that are limited to taking 1∼4 input images and can only reconstruct a small portion of a large scene, Long-LRM reconstructs the entire scene in a single feed-forward step. On large-scale scene datasets such as DL3DV-140 and Tanks and Temples, our method achieves performance comparable to optimization-based approaches while being two orders of magnitude more efficient. Project page: `https://longggglrm.github.io`

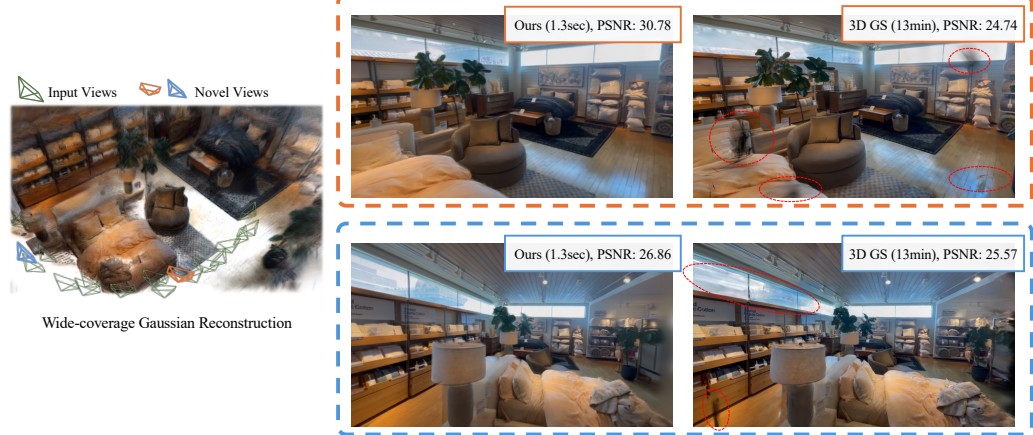

Figure 1: We introduce Long-LRM, a novel Gaussian reconstruction model capable of reconstructing a large real scene from a long sequence of up to 32 input images, with a wide viewing coverage at a resolution of $960 \times 540$, in just 1.3 seconds. Notably, as a feed-forward generalizable model, Long-LRM can achieve instant large-scale GS reconstruction with high rendering quality comparable to (and, as shown in the figure, sometimes even surpassing) the optimization-based 3D Gaussian splatting (3D GS), which requires over 13 minutes for optimization.

## 1 Introduction

3D reconstruction from multi-view images is a fundamental problem in computer vision, with applications ranging from 3D content creation, VR/AR, to autonomous driving and robotics. Recently, NeRF (Mildenhall et al., 2021) and various radiance field-based methods (Müller et al., 2022; Xu et al., 2022; Chen et al., 2022; Barron et al., 2023) have shown great potential in reconstructing high-quality 3D scenes from a set of posed images via differentiable rendering. However, these models are slow to reconstruct and not generalizable to unseen scenes, as they require optimization for each scene independently. While 3D Gaussian splatting (GS) (Kerbl et al., 2023) has significantly

advanced the reconstruction and rendering efficiency, it still typically requires at least 10 minutes to optimize for each scene and can not achieve an instant reconstruction.

Recently, generalizable 3D GS models (Szymanowicz et al., 2024; Tang et al., 2025) have been proposed to enable fast feed-forward GS reconstruction, avoiding per-scene optimization. Several methods (Charatan et al., 2024; Zhang et al., 2025; Liu et al., 2024a; Chen et al., 2025) have shown promising scene-level reconstruction results on real 3D captures by regressing per-pixel Gaussian primitives. In particular, GS-LRM (Zhang et al., 2025), following the principles of 3D large reconstruction models (LRMs) (Hong et al., 2024; Li et al., 2023; Wang et al., 2023) and leveraging a densely self-attention-based transformer (Vaswani, 2017) without using 3D inductive biases such as epipolar attention or sweeping volumes, has achieved state-of-the-art novel-view rendering quality on multiple challenging datasets. However, the previous generalizable GS models are designed to handle a small number of input images (typically 1-4) with limited viewing coverage, thus are incapable of reconstructing large real-world scenes, which require a wide view span and at least dozens of images. In such cases, per-scene optimization-based methods were still the only viable option.

Our goal is to enable fast and accurate GS reconstruction of large scenes with wide viewing coverage through direct feed-forward network prediction. To this end, we propose Long-LRM, a novel GS-based LRM that is able to *handle long-sequence input and achieve high-quality 3D GS reconstruction of large scenes from as many as 32 widely-displaced multi-view images at 960×540 resolution within only 1.3 seconds* on a single A100 80G GPU. As shown in Fig. 1, the photorealistic novel-view renderings produced by our approach has a quality comparable to or even better than 3D GS (Kerbl et al., 2023) that takes over 10 minutes for per-scene optimization.

Specifically, as inspired by GS-LRM, we patchify the multi-view input images into a sequence of patch tokens and consider the task of GS reconstruction as a sequence-to-sequence translation to regress pixel-aligned Gaussian primitives. However, unlike GS-LRM that focuses on 2-4 input images, our input setting with 32 960×540 images corresponds to an extremely long token sequence – **about 250K context length** (considering a patch size of $8{\times}8$) – which is highly challenging for dense transformers (as used by GS-LRM) due to their quadratic time complexity. Note that this length is even larger than many modern large language models (LLM), such as LLama3 (Dubey et al., 2024) with a context length of 128K.

To address this challenge, we leverage the recent advancements of state space models (SSMs) (Gu & Dao, 2023), designed to handle long-context reasoning efficiently with linear complexity. In particular, we propose a novel LRM architecture that combines Mamba2 (Dao & Gu, 2024) blocks with transformer blocks, enabling efficient sequential long-context reasoning while preserving critical global context. Additionally, we introduce a token merging module to further reduce the number of tokens in the middle of the network processing, along with a Gaussian pruning step to encourage efficient use of the dense per-pixel Gaussians. These combined designs allow us to train our Long-LRM using similar computational resources to GS-LRM, while successfully scaling up the input sequence length and achieving over $10\times$ faster training on long-sequence inputs, enabling fast, high-quality, wide-coverage reconstruction of large real scenes (see Tab. 3).

We train our Long-LRM on the recent DL3DV dataset (Ling et al., 2024), which comprises approximately 10K diverse indoor and outdoor scenes. We evaluate our model on both the DL3DV test set and the Tanks and Temples dataset (Knapitsch et al., 2017), using 32 input images for each scene. The results show that our direct feed-forward reconstruction achieves comparable novel view synthesis quality to the per-scene optimization results of 3D GS, while substantially reducing the reconstruction time – by two orders of magnitude (1.3 seconds vs. 13 minutes). **Our approach is the first feed-forward GS solution for wide-coverage scene-level reconstruction and the first to enable large-scale GS scene reconstruction in seconds.**

## 2 RELATED WORK

**3D Reconstruction.** Many traditional and learning-based 3D reconstruction methods have been focusing on pure geometry reconstruction, where surface meshes (Murez et al., 2020; Sun et al., 2021; Bozic et al., 2021; Stier et al., 2021) or depth maps (Zbontar & LeCun, 2016; Schönberger et al., 2016; Yao et al., 2018; Cheng et al., 2020; Kar et al., 2017; Duzceker et al., 2021; Sayed et al., 2022) are the target output. These methods usually involve explicit feature matching along the epipolar lines, followed by the prediction of TSDF or depth values performed by the neural networks. In

contrast, we adopt the recent 3D GS representation for joint geometry and appearance reconstruction, allowing for photo-realistic novel view synthesis.

**Neural reconstruction and rendering.** Instead of directly predicting the surface geometry, NeRF (Mildenhall et al., 2021) proposes to leverage differentiable volume rendering to regress novel-view images, supervised with a rendering loss. This implicit way of reconstruction eliminates the need for hard-to-obtain ground-truth 3D supervision while producing visually pleasing reconstruction results. However, NeRF reconstruction requires optimizing its network for each scene independently, taking hours or even days for reconstruction. Follow-up works have introduced advanced neural scene representations (Barron et al., 2023; Müller et al., 2022; Chen et al., 2022; Xu et al., 2022; Tancik et al., 2023; Barron et al., 2022), significantly improving time and memory efficiency. Among these, 3D Gaussian splatting (Kerbl et al., 2023) stands out for reducing reconstruction time to just dozens of minutes while maintaining high reconstruction quality and enabling real-time rendering. Variants of 3D GS, such as CityGaussian (Liu et al., 2025) and Octree-GS (Ren et al., 2024), further extend its capabilities to large-scale optimization and rendering. However, these methods are still unable to achieve instant prediction. We aim to build a scalable feed-forward reconstruction model, capable of achieving instant 3D GS reconstruction in seconds.

**Generalizable NeRF and 3D GS.** Previous attempts to develop generalizable NeRF models have primarily relied on classical projective geometric structures, such as epipolar lines (Yu et al., 2021; Wang et al., 2021; Liu et al., 2022; Suhail et al., 2022) or plane-sweep cost volumes (Chen et al., 2021; Johari et al., 2022; Lin et al., 2022; Zhang et al., 2022), to aggregate multi-view features from nearby views for local NeRF estimation. Recently, similar designs have been adapted to enable feed-forward scene-level 3D GS reconstruction with generalizable models (Charatan et al., 2024; Chen et al., 2025; Liu et al., 2024a). However, since both epipolar geometry and plane-sweep volumes depend on significant overlap between input views, these GS-based methods (as well as most prior NeRF-based methods) are limited to local reconstructions from a small number (1-4) of narrow-baseline inputs. On the other hand, GS-LRM (Zhang et al., 2025) avoids these 3D-specific structural designs and adopts an attention-based transformer, achieving state-of-the-art performance in this domain. However, GS-LRM still focuses on solving the problem of local reconstruction from just 2-4 views. In contrast, we incorporate Mamba (Gu & Dao, 2023; Dao & Gu, 2024) in our model architecture, enabling feed-forward GS reconstruction from 32 images, achieving complete large-scene reconstruction. Meanwhile, Gamba (Shen et al., 2024) and MVGamba (Yi et al., 2024) have recently utilized purely Mamba-based architectures for object-level GS reconstruction from 1-4 input views. Our model is instead a novel hybrid model that combines transformer and Mamba2 blocks, designed for long-sequence, high-resolution, scene-level reconstruction from up to 32 views.

**Efficient models for long sequences.** Transformer-based 3D large reconstruction models (LRMs) have emerged (Hong et al., 2024; Li et al., 2023; Xu et al., 2023; Wang et al., 2023; Wei et al., 2024; Xie et al., 2024; Zhang et al., 2025), enabling high-quality 3D reconstruction and rendering from sparse-view inputs. While transformers dominate various AI fields due to their flexibility with input modalities and scalability in model sizes, their quadratic time complexity makes them extremely slow when handling long sequences, often requiring thousands of GPUs for parallel computing (Dubey et al., 2024). Efficient architectures such as linear attention (Katharopoulos et al., 2020) and structured state space model (SSM) (Gu et al., 2021) have been proposed in NLP to deal with large corpus of text. Mamba (Gu & Dao, 2023), a variant of SSM, offers significant improvements by computing state parameters from each input in the sequence and has been successfully extended to tackle vision tasks (Zhu et al., 2024; Liu et al., 2024b; Lieber et al., 2024; Huang et al., 2024; Shen et al., 2024; Yi et al., 2024; Dong et al., 2024). Mamba2 (Dao & Gu, 2024) further restricts the state matrix $A$ and expands state dimensions, showing performance comparable to transformers on multiple language tasks. However, empirical studies (Waleffe et al., 2024) indicate that transformers still outperform Mamba2 in in-context learning and long-context reasoning—both critical for 3D reconstruction. Inspired by Waleffe et al. (2024) and Jamba (Lieber et al., 2024), we propose to apply a hybrid architecture combining transformer and Mamba2 blocks for long-sequence 3D GS reconstruction, achieving a balance between training efficiency and reconstruction quality (see Tab. 3).

## 3 METHOD

We present our Long-LRM method in this section. We give an overview in Sec. 3.1, the implementation details of the Mamba2 blocks in Sec. 3.2 and additional designs for memory reduction (e.g.,

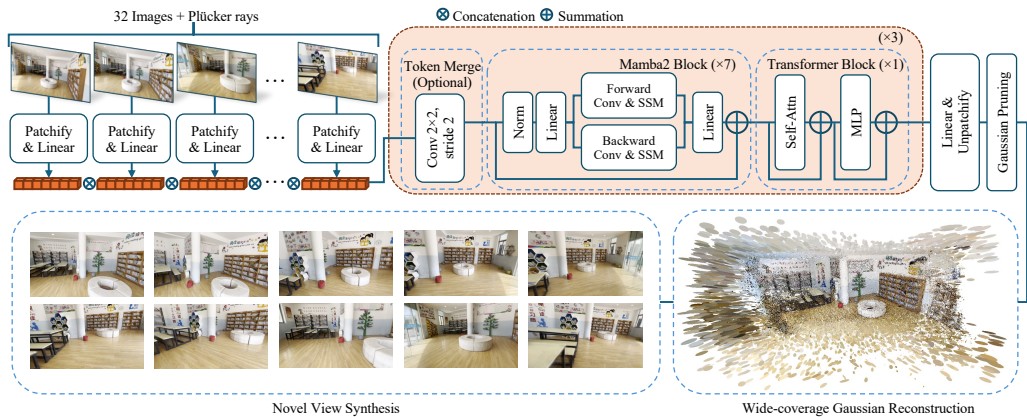

Figure 2: Long-LRM takes up to 32 input images along with their Plücker ray embeddings as model input, which are then patchified, linearly transformed, and concatenated into token sequences. These tokens are processed through an optional token merging module, followed by a sequence comprising Mamba2 blocks (×7) and a Transformer block (×1). This entire processing structure is repeated three times (×3) to ensure effective handling of the long-sequence inputs and comprehensive feature extraction. Fully processed, the tokens are unpatchified and decoded into Gaussian parameters, followed by Gaussian pruning to generate the final 3D GS representation. The bottom section of the figure illustrates the resulting novel view synthesis and wide-coverage Gaussian reconstruction, demonstrating Long-LRM's capability to handle extensive view coverage and produce high-quality, photorealistic reconstructions.

token merging) in Sec. 3.3. We end with a discussion of the training objectives in Sec. 3.4 that help the model to effectively converge.

## 3.1 OVERALL ARCHITECTURE

As shown in Fig. 2, we follow prior work (Xu et al., 2023; Wei et al., 2024; Zhang et al., 2025) to tokenize the channel-wise concatenated RGB images and Plücker rays. Similar to GS-LRM (Zhang et al., 2025), we view the per-pixel GS prediction as a sequence-to-sequence mapping. But crucially, we use a hybrid of Mamba2 blocks and transformer blocks, following the studies in Waleffe et al. (2024) and Lieber et al. (2024), for better scalability to higher resolution and denser views, while GS-LRM solely builds upon transformer blocks.

In our implementation, each hybrid block consists of 7 Mamba blocks and one transformer block, which we empirically observe to be a balanced configuration. For the transformer blocks, we use global self-attention, as done in recent LRMs (Wei et al., 2024; Zhang et al., 2025). We detail our implementation of Mamba2 blocks in Sec. 3.2. A token merging stage is optionally injected before the hybrid block to further speed up the processing, which is detailed later in Sec. 3.3.

We decode per-pixel Gaussian parameters from the output tokens in the same way as GS-LRM. But we apply additional training-time and test-time pruning of the extremely dense Gaussians to improve efficiency at high resolution and increased views.

## 3.2 MAMBA2 BLOCK

A Mamba block (Gu & Dao, 2023), similar to a transformer block, processes a token sequence of shape $L \times D$ by mixing the token information, and outputs a token sequence of the same shape. For a sequence of length $L$, transformer block has a computational complexity of $O(L^2)$ while Mamba effectively reduces it to $O(L)$. Thus, it is suitable for the dense reconstruction task in our Long-LRM.

Being a variant of SSM, Mamba at its core processes each input token $x$ by formula

$$h_t = \mathbf{A}h_{t-1} + \mathbf{B}x_t \tag{1}$$

$$y_t = \mathbf{C}h_t \tag{2}$$

where $h$ is the hidden state, $y$ is the output token, $t$ is the sequence index, and $\mathbf{A}$, $\mathbf{B}$, $\mathbf{C}$ are parameters. Different from previous work (Gu et al., 2021), Mamba computes $\mathbf{A}$, $\mathbf{B}$, $\mathbf{C}$ from the input with a linear

layer instead of storing them as model parameters. It's worth noting that similar to transformer block, Mamba block can be highly parallelized in terms of computation for leveraging the massive GPU compute power, which is one core factor driving its increasing popularity.

The novel Mamba2 (Dao & Gu, 2024) block improves over Mamba by further restricting the state matrix $\mathbf{A}$ to be a scalar times identity structure, allowing the usage of efficient block multiplication and expansion to larger state dimensions, showing performance comparable to transformers on multiple language tasks. However, since the Mamba2 block is designed for language tasks, it only scans through the tokens in one direction, which is suboptimal for images. Following Vision Mamba (Zhu et al., 2024), we take bi-directional scans over the concatenated token sequence. Specifically, we first compute the state parameters from the input using one linear layer; then we run the SSM block in both forward and backward directions on the token sequence. Finally, we sum up the output tokens from the two scans before going through another linear layer. We also did some preliminary exploration of more complex scan patterns as in VMamba (Liu et al., 2024b) and LocalMamba (Huang et al., 2024), but we observed a substantial decrease in speed, and hence we decided not to adopt them.

### 3.3 TOKEN MERGING AND GAUSSIAN PRUNING

Boosting up input view number and image resolution can drastically increase the token sequence length. With 32 960×540 images and patch size 8, the length can reaches about 260k, highly challenging even for linear-complexity models like Mamba. Empirically, we also find even the all-Mamba2 variant of our model runs out of memory under our highest resolution setting (see Tab. 3). To further reduce memory usage, we propose to merge the tokens in the middle of the model as well as to prune the Gaussians before rendering novel views.

Token merging achieves a fine-to-coarse effect similar to the traditional multi-level CNN encoders and effectively reduces token sequence length down to 1/4. We first reshape the token sequence from $L \times D$ back to $N \times \frac{H}{p} \times \frac{W}{p} \times D$ where $p$ is the original patch size. Then, we apply a channel-wise $2 \times 2$ 2D convolution with stride 2, resulting in output shape $N \times \frac{H}{2p} \times \frac{W}{2p} \times D'$, where $D'$ is the new token dimension that can differ from the original one. Finally, we reshape it back to $\frac{L}{4} \times D'$ where each token now has an 'effective' patch size of $2p$. In our ablation studies (Tab. 3), we find our token merging design does not sacrifice much reconstruction quality, while significantly reducing memory usage and increasing training speed.

Even with token merging, our per-pixel Gaussian prediction still brings us an enormous quantity of Gaussians at the end (∼17 million for 32 images with resolution 960×540), which is likely more than we need for a high-quality reconstruction due to the overlap between the input view frustums. To encourage the model to use a compact set of Gaussians, we apply a punishment on the opacity of all Gaussians (detailed in Sec. 3.4). With the effective reduction in the number of visible Gaussians, we can thus simply prune away a certain percentage of Gaussians with low opacity. Empirically, we find no difference in rendering quality if we prune away Gaussians with opacity below 0.001. Beside pruning during inference, we also apply the Gaussian pruning to the 960×540 resolution training. We keep fixed-number Gaussians instead of using opacity threshold to ensure near-constant training memory usage. O.w., the training can go out of memory for some scenes.

### 3.4 TRAINING OBJECTIVES

Lastly, we illustrate the training objectives for Long-LRM.

**Rendering loss.** Following previous work (Zhang et al., 2025), we use a combination of Mean Squared Error (MSE) loss and Perceptual loss

$$\mathcal{L}_{\text{image}} = \frac{1}{M} \sum_{i=1}^{M} \left( \text{MSE}\left(\mathbf{I}_i^{\text{gt}}, \mathbf{I}_i^{\text{pred}}\right) + \lambda \cdot \text{Perceptual}\left(\mathbf{I}_i^{\text{gt}}, \mathbf{I}_i^{\text{pred}}\right) \right) \tag{3}$$

to supervise the quality of the rendered images, where $\lambda$ is set to 0.5. While training solely with rendering loss can achieve competitive visual quality to our final model (see Sec. 5.2), we further introduce two regularization terms to improve training stability and inference efficiency.

**Depth regularization for training stability.** Training instability is a well-known curse for large-scale training. In our task, we observe that the instability comes from the difficulty of optimizing the

Gaussian positions. With rendering loss only, the model will produce ill-posed Gaussians known as "floaters", which does not lie on the actual 3D surface – a common issue for novel view synthesis (see the black "floaters" in Fig. 1). To stabilize training, we add a scale-invariant depth loss

$$\mathcal{L}_{\text{depth}} = \frac{1}{M} \sum_{i=1}^{M} \text{Smooth-L1} \left( \mathbf{D}_i^{\text{da}}, \mathbf{D}_i^{\text{pred}} \right) \qquad (4)$$

where $\mathbf{D}_i^{\text{da}}$ is the disparity map predicted by DepthAnything (Yang et al., 2024), and $\mathbf{D}_i^{\text{pred}}$ is the disparity map obtained from the predicted position of the per-pixel Gaussians. Following Yang et al. (2024), we normalize the disparity maps by subtracting their medians $t(d_i)$ and then dividing by their mean absolution deviation from the medians $\frac{1}{HW} \sum |d_i - t(d_i)|$. This soft depth supervision effectively helps reduce the chance of the training divergence.

**Opacity regularization for inference efficiency.** Since our per-pixel prediction strategy renders a dense set of Gaussians, to encourage an efficient use of the Gaussians, we apply a small L1 regularization on the opacity

$$\mathcal{L}_{\text{opacity}} = \frac{1}{N} \sum_{i=1}^{N} |o_i| \qquad (5)$$

where the opacity values are between 0 and 1. Intuitively, L1 can encourage the sparsity of the regularized terms (Tibshirani, 1996). We empirically observe that adding this loss can drastically push the percentage of Gaussians with opacity above 0.001 from 99% down to around 40% (see Tab. 5). With these near-zero opacity Gaussians, we can perform Gaussian pruning as discussed above in Sec. 3.3 and both reduce the Gaussian splatting loading time and increase the rendering speed for better model serving experience. This regularization also enables the extreme $960 \times 540$ resolution training where in-training pruning is used.

**Overall training loss.** Our total loss is thus the rendering loss and the weighted regularization loss terms discussed above:

$$\mathcal{L} = \mathcal{L}_{\text{image}} + \lambda_{\text{opacity}} \cdot \mathcal{L}_{\text{opacity}} + \lambda_{\text{depth}} \cdot \mathcal{L}_{\text{depth}} \qquad (6)$$

where we set $\lambda_{\text{opacity}} = 0.1$ and $\lambda_{\text{depth}} = 0.01$.

## 4 EXPERIMENTS

### 4.1 DATASETS

DL3DV (Ling et al., 2024) is a recently published large-scale, real-world scene dataset for 3D reconstruction and novel view synthesis. It features a diverse variety of scene types, with both indoor and outdoor captures. It consists of two parts: DL3DV-10K is the training split, consisting of 10,510 high-resolution videos, each accompanied by 200~300 keyframes with camera pose annotation (obtained from COLMAP (Schönberger et al., 2016)); DL3DV-140 Benchmark is the test split, containing 140 test scenes. We train our model on DL3DV-10K and evaluate on the DL3DV-140 Benchmark. We also perform zero-shot inference on Tanks and Temples (Knapitsch et al., 2017), another real-world scene dataset for novel view synthesis. It also contains 200~300 keyframes with camera pose annotation (obtained from COLMAP) for each scene. Following previous work (Kerbl et al., 2023; Liu et al., 2024a), we use the `train` and the `truck` scene from Tanks and Temples. In addition, a comparison with SOTA feed-forward GS methods under a sparse two-view setting is conducted on RealEstate10K (Zhou et al., 2018), a real-world indoor scene dataset, following the same train test split and evaluation setting introduced by pixelSplat Charatan et al. (2024).

### 4.2 IMPLEMENTATION AND EXPERIMENT DETAILS

**Architecture Details.** Our model consists of 24 blocks in total, with every 7 Mamba2 blocks followed by 1 transformer block, repeating 3 times. We start with patch size 8 and token dimension 256. We perform token merging at the beginning of the 9th block, with patch size expanded to 16 and token dimension expanded to 1024. For Mamba2 blocks, we use a state dimension 256, an expansion rate 2 and a head dimension 64. For transformer blocks, we use a head dimension 64 and an MLP dimension ratio of 4. We use the FlashAttentionV2 (Dao, 2024) implementation which optimizes the GPU IO utilization for long sequences.

**Training Settings.**    Directly training the model on high-resolution images is extremely inefficient; therefore we opt for an low-to-high-resolution curriculum training schedule, with three training stages, using image resolutions of $256 \times 256$, $512 \times 512$ and $960 \times 540$.

Specifically, in the 1st stage, training images are resized so the shorter side is 256 and then center-cropped to square. For the training view selection, we first randomly pick a consecutive subsequence ranging from 64 frames to 128 frames, then uniformly sample 32 images as input and sample 8 images as target. Input and target are sampled independently and thus they can overlap. We randomly shuffle the input view order with probability 0.5 and reverse the input view order also with probability 0.5. We train with a peak learning rate of $4\mathrm{E}{-}4$ and the AdamW optimizer (Loshchilov & Hutter, 2017) with a weight decay of $0.05$. The learning rate is linearly warmed up in the first 2K steps and then cosine decayed. We use a batch size of 256, and train for 60K steps.

In the 2nd stage, we resize and crop the images to $512 \times 512$, decrease the peak learning rate to $4\mathrm{E}{-}5$, and train the model for 10K steps at batch size 64. The view selection protocol remains the same.

In the last stage, we resize the images to $960 \times 540$ without square cropping, expand the view selection sampling range to the entire sequence (about 200~300 frames for DL3DV), and keep training the model for another 10K steps at batch size 64. We perform Gaussian pruning in this stage to save GPU memory usage, where we only keep top 40% of the Gaussians ranked by opacity plus 10% randomly sampled from the rest. We augment the FOV of the images by randomly center-cropping the images to 0.77~1.0 of the original size and resize back, in order to fit a broader range of camera models. We optionally finetune a model with 16 images as input.

**Evaluation Settings.**    During evaluation, our goal is to reconstruct the scene captured by the entire video sequence. Following previous work (Barron et al., 2022; Kerbl et al., 2023), we uniformly pick every 8-th image of the sequence as the test split. From the rest of the sequence, we use $K$-means clustering (based on camera positions and directions) for choosing the input views to ensure the coverage of the scene. The number of clusters is set of the number of input views. We simply choose the cameras closest to the cluster centers as the input split. We use an image resolution of $960 \times 540$ during the evaluation. We perform Gaussian pruning during evaluation by only keeping the top 50% of the Gaussians with highest opacity values, where 50% is a safe range with negligible quality loss.

## 4.3   RESULTS

| Input Views | Method | Time↓ | DL3DV-140 | | | Tanks&Temples | | |
|---|---|---|---|---|---|---|---|---|
| | | | PSNR↑ | SSIM↑ | LPIPS↓ | PSNR↑ | SSIM↑ | LPIPS↓ |
| 16 | 3D GS$_{30k}$ | 13min | 21.20 | 0.708 | **0.264** | 16.76 | **0.598** | **0.334** |
| | Ours | **0.7sec** | **22.66** | **0.740** | 0.292 | **17.51** | 0.555 | 0.408 |
| 32 | 3D GS$_{30k}$ | 13min | 23.60 | 0.779 | **0.213** | 18.10 | **0.688** | **0.269** |
| | Ours | **1.3sec** | **24.10** | **0.783** | 0.254 | **18.38** | 0.601 | 0.363 |

Table 1: **Quantitative comparison to 3D Gaussian splatting optimization.** 'Time' refers to the total inference/optimization time for each scene. The image resolution is $960 \times 540$.

| Method | PSNR↑ | SSIM↑ | LPIPS↓ |
|---|---|---|---|
| pixelSplat | 25.89 | 0.858 | 0.142 |
| MVSplat | 26.39 | 0.869 | 0.128 |
| GS-LRM | 28.10 | 0.892 | 0.114 |
| Ours (w/ TM) | 27.26 | 0.872 | 0.130 |
| Ours (w/o TM) | **28.44** | **0.893** | **0.113** |

Table 2: **Quantitative comparison on RealEstate10K under 2-view setting.** 'TM' refers to token merging. The image resolution is $256 \times 256$.

Our approach achieves wide-coverage, scene-level 3D Gaussian splatting reconstruction from up to 32 high-resolution input images, which, to the best of our knowledge, no other method can accomplish. Recent works like pixelSplat (Charatan et al., 2024), MVSplat (Chen et al., 2025), MVSGaussian (Liu et al., 2024a), and GS-LRM (Zhang et al., 2025) are limited to processing 1–4 input images, with pixelSplat and MVSplat showing results only at $256 \times 256$ resolution. Most of these methods rely on traditional 3D inductive biases, such as epipolar projection and cost volumes, which are suited for narrow-view inputs with large overlaps but struggle with wide-coverage, high-resolution settings. Moreover, naively extending these methods to handle more input views and higher resolutions leads to out-of-memory issues and requires significant architectural changes. Therefore, we compare our method with the original optimization-based 3D Gaussian splatting in the high-resolution, wide-coverage setting on the DL3DV and Tanks&Temples datasets, and also compare with previous feed-forward methods in the low-resolution two-input setting on the RealEstate10k dataset.

**High-resolution, wide-coverage reconstruction.** In Table 1, we show the quantitative comparison results with the optimization-based 3D GS on two real-world large-scene datasets: DL3DV-140 Benchmark (Ling et al., 2024) and Tanks and Temples (Knapitsch et al., 2017). We show results under the sparser 16 input-view setting as well as the 32 input-view setting. Our model is capable of reconstructing an unseen novel scene from long-sequence input in a feed-forward manner within as little time as 1.3 seconds, $600\times$ faster than 3D GS optimization (13 minutes for 30K steps). Reconstruction quality-wise, our feed-forward reconstruction results are comparable with 3D GS with 30K optimization steps. Our model takes lead in terms of PSNR (with the gap larger in the sparser 16-view setting: +1.2 for DL3DV-140 and +0.4 for Tanks and Temples), while 3D GS performs better in terms of LPIPS. We speculate this is because 3D GS optimization is much stronger at directly "copying" the input images into the reconstructed scene with these many optimization steps, and thus can render images with local color distribution extremely similar to the test images. However, without any prior knowledge in 3D geometry, it can easily overfit to the input views when the input is sparse. As demonstrated in our qualitative comparisons with 3D GS (Fig. 1 and 3), Long-LRM shows significant improvements in reducing floater artifacts. This improvement can be attributed to two key factors. First, as a feed-forward method, Long-LRM leverages prior knowledge distilled from a large training dataset, helping to avoid floaters in unseen views. Second, we have adopted regularization terms like the opacity loss and the soft depth supervision, which are effective in mitigating floater artifacts. More visualization and interactive results can be found on our website and in Appendix.

**Low-resolution, sparse-view reconstruction.** In Table 2, we present a quantitative comparison with state-of-the-art feed-forward GS methods on the RealEstate10K dataset at a $256\times256$ resolution with 2 input views, a setting commonly used in prior works. Our Long-LRM, without token merging, achieves the best overall quality, outperforming pixelSplat and MVSplat by a large margin of over 2dB PSNR and slightly surpassing the transformer-based GS-LRM, highlighting the effectiveness of our hybrid model. While adding token merging slightly reduces quality in this sparse-view setting, it still achieves competitive results, surpassing pixelSplat and MVSplat. Importantly, token merging enables Long-LRM to handle higher resolutions and longer sequences, effectively addressing the scalability challenges that are central to our work. Overall, our approach not only leads to state-of-the-art rendering quality in the classical sparse-view setting but also enables wide-coverage, high-resolution, large-scene reconstruction that other feed-forward methods cannot achieve.

## 5 ANALYSIS

### 5.1 ABLATION STUDIES OF MODEL DESIGNS

| Input Views | Image Size | Batch Size / GPU | Train Step | Block Type | Token Merge | Patch Size | Token Dimension | #Param | Iteration Time (sec) | GPU Memory (GB) | PSNR↑ |
|---|---|---|---|---|---|---|---|---|---|---|---|
| 4 | 256 | 16 | 100K | Transformer (GS-LRM) | / | 8 | 1024 | 327M | 2.3 | 44 | 21.13 |
| | | | | Mamba2 | / | 8 | 1024 | 190M | 2.8 | 35 | 19.82 |
| | | | | {7M1T}$\times$3 | / | 8 | 1024 | 206M | 2.6 | 35 | **21.58** |
| | | | | {7M1T}$\times$3 | @9 | 8 $\rightarrow$16 | 256 $\rightarrow$1024 | **162M** | **1.9** | **20** | 21.25 |
| 32 | 256 | 4 | 60K | Transformer (GS-LRM) | / | 8 | 1024 | 327M | 14.5 | 68 | too slow |
| | | | | Mamba2 | / | 8 | 1024 | 190M | 6.0 | 70 | 24.28 |
| | | | | {7M1T}$\times$3 | / | 8 | 1024 | 206M | 7.1 | 70 | **26.82** |
| | | | | {7M1T}$\times$3 | @9 | 8 $\rightarrow$16 | 256 $\rightarrow$1024 | **162M** | **3.5** | **25** | 25.62 |
| 32 | 512 | 1 | 10K$^*$ | Transformer (GS-LRM) | / | 8 | 1024 | 327M | 50.5 | 44 | too slow |
| | | | | Mamba2 | / | 8 | 1024 | 190M | 7.4 | 62 | 24.83 |
| | | | | {7M1T}$\times$3 | / | 8 | 1024 | 206M | 11.5 | 64 | **28.16** |
| | | | | {7M1T}$\times$3 | @9 | 8 $\rightarrow$16 | 256 $\rightarrow$1024 | **162M** | **4.0** | **23** | 27.46 |
| 32 | 960$\times$540 | 1 | 10K$^*$ | **All other variants are out of memory.** | | | | | | | |
| | | | | {7M1T}$\times$3 | @9 | 8 $\rightarrow$16 | 256 $\rightarrow$1024 | 162M | 12.6 | 53 | 27.32 |

Table 3: **Ablation studies on model architecture.** We study how the model architecture affects training time and memory efficiency as well as the reconstruction quality. All variants have 24 blocks in total. **{7M1T}$\times$3** refers to our "7 Mamba2 blocks + 1 Transformer block, repeating 3 times" model architecture. **@9** means the token merging happens at the beginning of the 9th block. Models are trained on DL3DV-10K and evaluated on DL3DV-140 Benchmark. $^*$The 512-resolution models are finetuned from the checkpoints of their 256-resolution counterparts, and the 960-resolution from the 512-resolution checkpoints.

We study how the model architecture variants scale with long input image sequence for both training efficiency and reconstruction quality. As shown in Table 3, we consider 4 experimental setups with different sequence lengths: 1. sparse low-resolution ('Input Views'=4, 'Image Size'=256), 2. dense low-resolution ('Input Views'=32, 'Image Size'=256), 3. dense high-resolution ('Input Views'=32, 'Image Size'=512), 4. dense ultra-resolution ('Input Views'=32, 'Image Size'=960×540) [1]. The results of different model architecture under the same setup are presented within a Table block (i.e., every four rows). We study four model variants: all transformer blocks (row 1; equivalent to GS-LRM), all Mamba2 blocks (row 2), hybrid blocks but without token merging (row 3), and hybrid blocks with token merging (row 4; our final model). All variants have 24 blocks in total. We illustrate the number of model parameters ('#Param'), the training iteration time, the GPU memory usage, and PSNR reconstruction metric in the last four columns. The detailed experimental setup of this ablation study can be found in Appendix. We next highlight the key observations.

**Comparisons to Transformer.** Transformer's performance is comparable to our model under the 4-view 256-resolution (the 1st block in Tab. 3) setting. However, its training speed explodes for larger visual inputs, either with dense view or high resolutions. In our 3rd experiment setup (32 views with resolution 512), the per-iteration training time with batch size=1 can go up to 50.5 seconds, which is unaffordable to train. This is due to the quadratic time complexity of transformers.

**Comparisons to Mamba2.** The Mamba2 variant shows a more manageable increase in time as the input scales up but leads to a noticeable decline in reconstruction quality compared to other variants. For instance, in the 256-resolution, 4-view setting (1st block in Tab 3 ), the Mamba2 variant exhibits a 1.8 PSNR drop compared to our hybrid model (row 3). This performance gap widens with longer sequences, reaching 2.5 PSNR for 32 views (2nd block in Tab 3) and 3.3 PSNR at 512 resolution (3rd block in Tab 3). This decrease in quality is possibly due to Mamba's purely state-based design, which struggles to capture long-range dependencies effectively.

**Effectiveness of Token Merging.** Comparing with transformer and Mamba2, our hybrid variant (third row in each block) gets the best of both worlds – the reconstruction quality (in terms of 'PSNR') comparable to transformer and the speed (in terms of 'Iteration Time') comparable to Mamba2. On top of it, with the token merging design (last row in each block), our final model successfully reduces both time and memory usage down to $1/3$ in the $512 \times 512$ setting, without sacrificing too much reconstruction quality. Token merging with Gaussian pruning also further enables scaling up to $960 \times 540$ resolution with stable reconstruction, where all other variants are out-of-memory.

## 5.2 ABLATION STUDIES OF TRAINING OBJECTIVES

| Input Views | Image Size | Loss Type | PSNR↑ | % Gaussians w/ opacity>0.001 |
|---|---|---|---|---|
| | | rendering-only | 20.43 | 99.2 |
| 4 | 256 | +opacity | 20.96 | 68.3 |
| | | +opacity+depth | 21.25 | 70.1 |

Table 4: **Ablation studies on training objectives.** We study how the opacity loss and the depth supervision affect the reconstruction quality as well as the Gaussian usage.

| Input Views | Image Size | Input Sampling Range (frame) | w/ opacity loss | % Gaussians w/ opacity>0.001 |
|---|---|---|---|---|
| 4 | 256×256 | 16 | ✗ | 99.2 |
| 4 | 256×256 | 16 | ✓ | 68.3 |
| 32 | 256×256 | 64 ∼ 128 | ✓ | 41.8 |
| 32 | 512×512 | 64 ∼ 128 | ✓ | 34.1 |
| 32 | 960×540 | 200 ∼ 300 | ✓ | 33.3 |

Table 5: **Gaussian usage impacted by opacity loss and input size**.

**Impact of the regularization terms.** In Tab. 4, we show the impact of the two regularization terms introduced in Sec. 3.4: the opacity loss and the depth supervision. From the table, we see that adding the opacity loss can significantly reduce the number of visible Gaussians (% of Gaussians with opacity above 0.001), while having negligible impact on model rendering performance. The depth supervision help improve the rendering quality by guiding the "floater" Gaussians to the position of the true surfaces. We observe it also slightly lifts the number of visible Gaussians, which is reasonable because now the model can drive the "floater" Gaussians to their correct positions instead of simply deleting them by assigning them low opacity values. Also due to this, training with depth supervision significantly reduces the chance of gradient explosions in our experiments.

---

[1] Note that here the terminology of 'sparse', 'dense', 'low', 'high', 'ultra' are all relative. We use these terminology for simplicity and clarity.

**Impact of opacity loss on Gaussian Usage.** In Tab. 5, we show how the opacity loss and the input size affects the Gaussian usage (percentage of Gaussians with opcacity $> 0.001$). Comparing row-1 and row-2, we observe that the opacity regularization loss introduced in Sec. 3.4 can effectively reduce the number of 'high'-opacity Gaussians shown in the last column. Furthermore, our model learns to adaptively use a different number of Gaussians when the input varies. As the image resolution increases (hence more per-pixel Gaussians predicted), the chance that multiple pixels can be covered by the same Gaussian increases as well, and thus the percentage of Gaussian usage decreases. However, as the sampling range (i.e., the maximum difference in frame indices of the input views, shown as 'Input Sampling Range') increases, the overlap between input views decreases, and thus the model needs to retain more Gaussians to keep reconstruction quality, resulting in negligible drop in Gaussian usage in the last row.

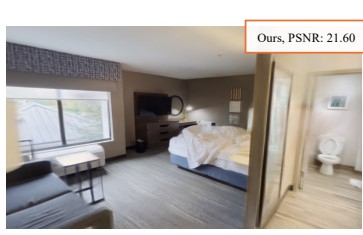
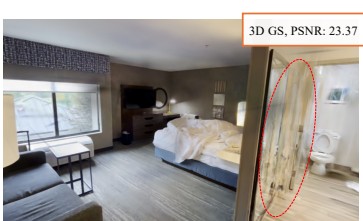
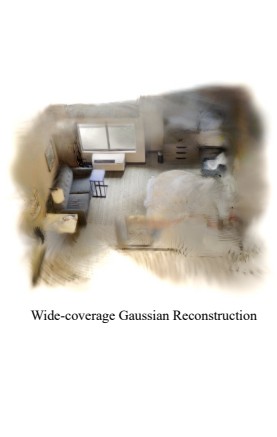
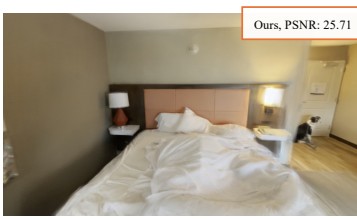
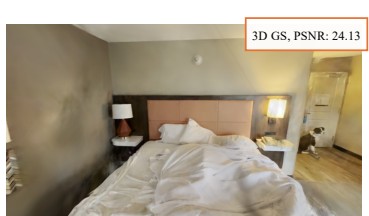

Wide-coverage Gaussian Reconstruction

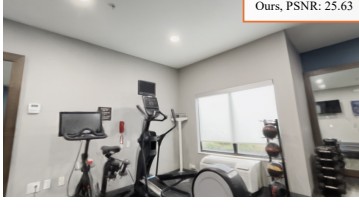
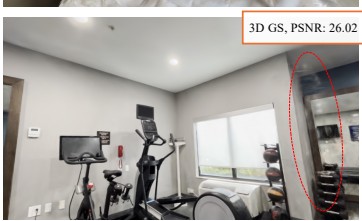
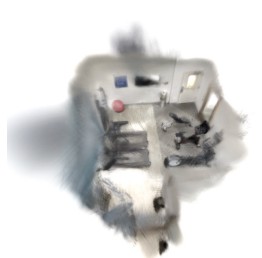
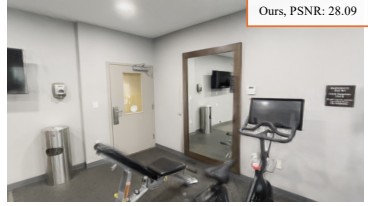
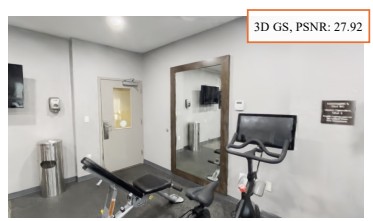

Wide-coverage Gaussian Reconstruction

Figure 3: Qualitative comparisons between Long-LRM and 3D GS, reconstructed from 32 input images at $960 \times 540$ resolution. The left two columns show our wide-coverage Gaussian reconstruction, while the right column shows results from 3D GS. Our approach maintains high-quality reconstruction with competitive or even superior PSNR values, demonstrating the ability to generate accurate details and fewer artifacts in challenging regions. The red ellipses highlight areas where 3D GS struggles with artifacts or inaccuracies, whereas Long-LRM produces cleaner and more photorealistic outputs.

## 6 CONCLUSIONS

In this work, we introduce Long-LRM, a novel model for fast and scalable 3D Gaussian splatting reconstruction. By combining Mamba2 and transformer blocks, along with token merging and Gaussian pruning, Long-LRM can instantly reconstruct a wide-coverage 3D GS scene from 32 images at a high resolution of $960 \times 540$ in just 1.3 seconds, leading to high rendering quality comparable to optimization-based methods such as 3D Gaussian splatting. Our approach is the first feed-forward GS solution for wide-coverage scene-level reconstruction.

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

Figure 4: Demonstration of our Long-LRM's novel view synthesis capabilities. The left column illustrates the wide-coverage Gaussian reconstruction achieved by our model, while the right columns show high-quality synthesized novel views from different perspectives. These examples demonstrate Long-LRM's ability to handle diverse and complex scenes, accurately reconstructing fine-level details, and generating photorealistic views from multiple angles, effectively capturing both geometric and appearance variations across different scenes.

## A    MORE QUALITATIVE RESULTS

We show more qualitative results of our Long-LRM on large-scale scenes using 32 wide-coverage input views at $960 \times 540$ image resolution in Fig. 4. For more visual results with rendered long-trajectory videos, please refer to our project webpage (`https://longggggglrm.github.io`).

## B    EXPERIMENTAL DETAILS FOR MODEL ARCHITECTURE ABLATION STUDIES

In Table 3, we present the model architecture ablation studies with different length of input sizes. We train all variants on DL3DV-10K and evaluate on DL3DV-140. The number of training steps are empirically decided based on the model convergence, and set to be the same. We study the model behavior under four different settings: 4 input views at $256 \times 256$, 32 input views at $256 \times 256$, and 32 input views at $512 \times 512$, and our extreme setting: 32 input views at $960 \times 540$.

For these ablation studies, we use a shorter frame range during evaluation for fair comparisons among each experiments. In details, we choose the first 96 frames from the original video frame sequence, then uniformly sample 8 test views. The training 4 to 32 training views are then uniformly sampled from the rest views, i.e., not overlapping to the testing views. We kept the same set of training and testing views for different experimental setups. The input images are resized and center-cropped to squares except for the last row.

## C    ADDITIONAL EXPERIMENT RESULTS

**Comparison with other 3D GS variants.** We show comparison with 3D GS and two of its variants, Scaffold-GS and Mip-Splatting, in Table 6. In particular, under the same input setting, Mip-Splatting achieves similar performance to 3D GS while Scaffold-GS leads to superior quality. It is important to

note that the contributions of these works are orthogonal to ours. Our work focuses on large-scale feed-forward GS, addressing challenges in scalability and efficiency. We leverage the original 3D GS representation in our model. In contrast, Mip-Splatting and Scaffold-GS focus on improving the 3D GS representations, instead of developing feed-forward solutions. In particular, Mip-Splatting emphasizes anti-aliasing during rendering, while Scaffold-GS focuses on regularizing the positions of Gaussians during optimization. Our approach could potentially be extended to incorporate these advanced representations, which we leave as a direction for future research.

| Input Views | Method | Feed-Forward | Time↓ | DL3DV-140 PSNR↑ | DL3DV-140 SSIM↑ | DL3DV-140 LPIPS↓ | Tanks&Temples PSNR↑ | Tanks&Temples SSIM↑ | Tanks&Temples LPIPS↓ |
|---|---|---|---|---|---|---|---|---|---|
| 16 | 3D GS$_{30k}$ | ✗ | 13min | 21.20 | 0.708 | 0.264 | 16.76 | 0.598 | 0.334 |
|  | Mip-Splatting$_{30k}$ | ✗ | 13min | 20.88 | 0.712 | 0.274 | 16.82 | 0.616 | 0.332 |
|  | Scaffold-GS$_{30k}$ | ✗ | 16min | 22.13 | 0.738 | **0.250** | 17.02 | **0.634** | **0.321** |
|  | Ours | ✓ | **0.7sec** | **22.66** | **0.740** | 0.292 | **17.51** | 0.555 | 0.408 |
| 32 | 3D GS$_{30k}$ | ✗ | 13min | 23.60 | 0.779 | 0.213 | 18.10 | 0.688 | 0.269 |
|  | Mip-Splatting$_{30k}$ | ✗ | 13min | 23.32 | 0.784 | 0.217 | 18.39 | 0.700 | 0.262 |
|  | Scaffold-GS$_{30k}$ | ✗ | 16min | **24.97** | **0.816** | **0.188** | **18.92** | **0.728** | **0.242** |
|  | Ours | ✓ | 1.3sec | 24.10 | 0.783 | 0.254 | 18.38 | 0.601 | 0.363 |

Table 6: **Quantitative comparison with per-scene optimization-based GS methods.** 'Feed-forward' column indicates whether the method performs zero-shot feed-forward prediction. 'Time' refers to the total inference/optimization time for each scene. **First place** is in bold, and second place is underlined. The image resolution is $960 \times 540$.

## D  LIMITATIONS

We now briefly discuss the limitations. While we successfully scaled the model to support 32 high-resolution views and achieved wide-coverage large-scale GS reconstruction, we observed only marginal performance improvements when further increasing the number of input views. Specifically, increasing the input to 64 views only lead to less than 1 dB PSNR improvement. Notably, 64 high-res images correspond to extremely long sequences, exceeding 500k in context length, which presents a significant challenge for current sequence processing models. Addressing this limitation will require future work to better manage ultra-long sequences. Additionally, since the entire DL3DV training set contains images with a fixed wide field of view (FOV), we found that our model struggles to generalize on test sets with significant FOV variations (e.g., the MipNeRF360 dataset with a much smaller FOV). We suspect this limitation is due to the use of Mamba2 blocks, as differing FOVs can alter the meaning of tokens at different positions. Developing models that can generalize effectively across varying FOVs may require more diverse datasets with a range of various FOVs, at a scale similar to DL3DV.

