# OpenReview forum: "Long-LRM: Long-sequence Large Reconstruction Model for Wide-coverage Gaussian Splats"
_ICLR.cc/2025/Conference — Submitted to ICLR 2025_

### Official Review · Reviewer_K9N5 · 2024-10-28

**Soundness:** 3
**Presentation:** 3
**Contribution:** 2
**Rating:** 5
**Confidence:** 4

**Summary:**

They propose a generalizable 3D Gaussian reconstruction model that can reconstruct a wide -coverage scene from a long sequence of input images with Mamba2 blocks. Some validation results shows the effeciveness compared with original 3DGS.

**Strengths:**

1.Introducing a method able to infer the 3DGS for wide-coverage scenes.

2.Utilize Mamba2 architecture to model the long token relations.

**Weaknesses:**

1.The comparison is not enough. Only compared with naive 3DGS. There are recent 3DGS/NeRF variants designed for large scale scene modeling: Zip-NeRF: Anti-Aliased Grid-Based Neural Radiance Fields, Scaffold-GS: Structured 3D Gaussians for View-Adaptive Rendering, Mip-Splatting: Alias-free 3D Gaussian Splatting.

2.Despite of the inference speed, it shows in the videos the floaters appear without further regularizations.

3.They main contribution is to use Mamba2 for long sequence modeling, which limits the technical contribution of the paper.

4.It is better to show the NVS comparison under sparse view setting compared with generalizable 3DGS methods like MVsplat and pixelSplat.

5.The method only shows the NVS results, it would be better to show some surface reconstruction results since with the development of "2DGS" and "High-quality Surface Reconstruction using Gaussian Surfels". Nowadays surface reconstruction with Gaussians already achieves very good results.

6.No regularization for aliasing effect is proposed.

**Questions:**

1.Is it able to seamlessly concat all sets of gaussians inferred by your models for large scale scenes which need hundreds of images? If so, how is it compared with city-scale reconstruction methods like CityGaussian and Octree-GS: Towards Consistent Real-time Rendering with LOD-Structured 3D Gaussians, which are designed for real large scale scene reconstruciton.

---

> ### Author Response · Authors · 2024-11-22
>
> 1. Comparison with recent 3DGS/NeRF variants, such as Zip-NeRF, Scaffold-GS, and Mip-Splatting.
>
>
> **Ans**: We include a comparison table with Scaffold-GS and Mip-Splatting below, omitting Zip-NeRF. Although Zip-NeRF achieves higher rendering quality, it requires dozens of hours to optimize a single scene, which is an order of magnitude slower than GS methods that typically take around 10+ minutes. In particular, under the same input setting, Mip-Splatting achieves similar performance to 3D GS while Scaffold-GS leads to superior quality.
>
> It is important to note that the contributions of these works are orthogonal to ours. Our work focuses on large-scale feed-forward GS, addressing challenges in scalability and efficiency. We leverage the original 3D GS representation in our model. In contrast, Mip-Splatting and Scaffold-GS focus on improving the 3D GS representations, instead of developing feed-forward solutions. Our approach could potentially be extended to incorporate these advanced representations, which we leave as a direction for future research.
>
> **Rendering quality on DL3DV-140 with 32-view input**
> | Method | Time | PSNR(↑) | SSIM(↑) | LPIPS(↓) |
> | -------- | :-------: | :-------: | :-------: | :-------: |
> | 3D GS | 13min | 23.60 | 0.779 | 0.213 |
> | Mip-Splatting | 13min | 23.32 |  0.784 | 0.217  |
> | Scaffold-GS | 16min | 24.97 | 0.816 | 0.188 |
> | Long-LRM | 1.3sec | 24.10 | 0.783 | 0.254 |
>
>
> (The number for Long-LRM slightly improves over the number reported in the original submission because, during the development of Long-LRM, only the first 7K scenes of the DL3DV10K dataset were available for training. The updated numbers, presented here, come from a version of Long-LRM trained on the full 10K scenes.)
>
>
>
> 2. Video results show floaters, suggesting the need for further regularization.
>
> **Ans**: We address the challenging task of wide-coverage scene reconstruction from 32 input images, a problem made difficult by the relatively limited view distribution in large scene spaces. Usually, standard per-scene optimization-based NeRF and 3D GS methods would require hundreds of input views to reconstruct such a large scene space. In our setting of 32 input views, even the standard per-scene optimization-based 3D GS can result in floater artifacts. However, as demonstrated in our qualitative comparisons with 3D GS in the paper, Long-LRM shows significant improvements in reducing floaters. This improvement can be attributed to two key factors. First, as a feed-forward method, Long-LRM leverages prior knowledge distilled from a large training dataset, helping to avoid floaters in unseen views. Second, we have already adopted regularization terms like the opacity loss and the soft depth supervision, which are effective in mitigating floater artifacts.
>
> Overall, wide-coverage feed-forward scene reconstruction remains a relatively new and underexplored problem. Our work makes a step by scaling feed-forward models to support 32 input images to address this problem. We look forward to future advancements that further scale feed-forward models to handle even more input views—potentially several hundreds—which could further reduce floaters and improve reconstruction quality.

---

> ### Author Response · Authors · 2024-11-22
>
> 3. Technical novelty.
>
> **Ans**: We emphasize that our work targets a **new and under-explored task**: achieving **feed-forward, wide-coverage 3D scene reconstruction from a large number of input images**. Prior to our work, large scene-level NeRF or 3DGS reconstructions could only be achieved by per-scene optimization methods. Recent feed-forward methods, such as GS-LRM, Gamba, and MVGamba, have made progress but focus on object-level reconstruction or sparse input views, leaving wide-coverage scene reconstruction largely unexplored. Our work is the first to demonstrate that feed-forward models can be effectively designed to handle large-scale scene reconstruction tasks, which we believe provides valuable insights to the community.
>
> A key aspect of our contribution is addressing the challenge of **long-sequence inputs** with up to ~250K tokens—a scale more than ten times larger than existing transformer- or Mamba-based 3DGS models. This significantly longer sequence introduces unique challenges in maintaining high reconstruction quality while managing computational efficiency.
>
> To tackle these challenges, we propose a novel **hybrid architecture** that combines transformers and Mamba2 blocks, achieving a balance between quality and efficiency. While similar hybrid architectures have been applied to other vision tasks, **we are the first to adopt this hybrid approach for 3D reconstruction**. In addition, we introduce **token merging** and **Gaussian pruning techniques** to further reduce computational complexity, enabling our method to handle significantly longer sequences compared to prior methods.
>
> We also emphasize that training at this large input scale is inherently more challenging and unstable than training on shorter sequences. To address this, we designed a multi-resolution training strategy with combined loss functions, enabling stable training at large scales. To the best of our knowledge, **this is the first feed-forward 3D reconstruction solution demonstrated at this large input scale with up to 32 images at a resolution of 960x540**. Furthermore, we conducted a comprehensive study and analysis of various design choices at the same large input scale; we hope this provides helpful insights into the trade-offs and performance of different techniques for long-sequence reconstruction tasks.
>
> We hope our work inspires the community to develop new technologies to address the challenges of the under-explored task of feed-forward wide-coverage 3D scene reconstruction.
>
>
> 4. Sparse-view NVS comparison with generalizable 3DGS methods like MVSplat and pixelSplat.
>
> **Ans**: As shown in the table below, Long-LRM outperforms all previous cost volume-based feed-forward GS methods under sparse settings and performs on par with GS-LRM without the token merging step. The token merging introduces a trade-off between scalability and model capacity. While it slightly reduces quality in this sparse-view setting, it enables Long-LRM to handle higher resolutions and longer sequences, addressing scalability challenges effectively.
>
> **Rendering quality on RealEstate10k with 2-view input**
> | Method | PSNR(↑) | SSIM(↑) | LPIPS(↓) |
> | -------- | :-------: | :-------: | :-------: |
> | pixelSplat | 25.89 | 0.858 | 0.142 |
> | MVSplat | 26.39 | 0.869 | 0.128 |
> | Long-LRM (w/ token merge) | 27.26 | 0.872 | 0.130 |
> | GS-LRM | 28.10 | 0.892 | 0.114 |
> | Long-LRM (w/o token merge) | 28.44 | 0.893 | 0.113 |
>
>
> 5. Surface reconstruction results should be included.
>
> **Ans**: We have updated our project page to include a video interleaving the color reconstruction and the geometry reconstruction. Please find this video at the top of the page (https://longgggglrm.github.io/).
>
>
> 6. Regularization for aliasing effects.
>
> **Ans**: Our work focuses on feed-forward wide-coverage 3D GS scene reconstruction, and addressing anti-aliasing is orthogonal to the scope of this study. It is worth noting that none of the existing feed-forward 3D GS methods (e.g. pixelSplat, MVSplat, GS-LRM) address the aliasing effect either. Extending feed-forward models to handle aliasing is indeed an interesting and valuable future direction, and we look forward to seeing advancements in this area.

---

> > ### Comment · Reviewer_K9N5 · 2024-11-25
> > **Reply to author's rebuttal**
> >
> > Thank you for expanding the experimental comparisons with Scaffold-GS, Mip-Splatting, GS-LRM and cost volume-based methods. While the additional evaluations are valuable, I have remaining concerns about the core technical contributions:
> >
> > Given that Scaffold-GS can already optimize high-quality scenes in reasonable time, the value proposition of Long-LRM is unclear, especially since it:
> >
> > 1.Produces lower quality results
> > Still requires 1.3 seconds per frame (not achieving real-time performance)
> >
> > 2.The technical novelty appears limited, as the combination of Mamba architecture with feed-forward Gaussian Splatting represents an incremental advance rather than a fundamental innovation.
> >
> > I have a more question: whether the proposed fast inference results can be the initialization of the SOTA 3DGS such as Scaffold-GS, so that the final optimization time and quality are improved?

---

> > > ### Author Response · Authors · 2024-11-27
> > >
> > > Thank you for your prompt response!
> > >
> > > 1. Comparison with Scaffold-GS
> > >
> > > **We would like to clarify that the time we reported is NOT per frame, but per scene. Specifically, Long-LRM takes 1.3 seconds for inference on 32 images (41ms / frame), and 0.7 seconds on 16 images (44ms / frame). Hence, our method works at real-time speeds.**
> > >
> > > As we have already emphasized, the contribution of Scaffold-GS is orthogonal to ours. Our primary focus is on exploring the feasibility of large-scale, feed-forward Gaussian splatting (GS), while Scaffold-GS enhances the representation of 3D GS by introducing a hierarchical structure for Gaussians. These advancements are complementary, and integrating 3D GS variants like Scaffold-GS into a feed-forward framework is an exciting avenue for future work. However, establishing a strong starting point, as we have done with our work, is a necessary precursor to such developments.
> > >
> > > Our feed-forward approach, which achieves a per-scene reconstruction time of just 1.3 seconds, is approximately **700 times** faster than Scaffold-GS, with only a 0.8 dB difference in reconstruction quality. To put this into context, reconstructing all 140 scenes in the DL3DV-140 dataset would take only **3 minutes** with Long-LRM on a single A100 GPU, compared to **30 hours** for Scaffold-GS or any other optimization-based 3D GS methods. We believe that our approach has already substantially improved the speed of scene reconstruction by 2 orders of magnitude with minimal performance loss w.r.t. state-of-the-art.
> > >
> > >
> > > 2. Technical Novelty
> > >
> > > We argue that Long-LRM is not just a simple concatenation of Mamba and 3D GS. Long-LRM is designed to process **250K tokens** from 32 high-resolution input images, a token length that is considered very long for even many modern LLMs. This scale introduces unique challenges that previous object-level feed-forward GS methods, such as Gamba, did not encounter.
> > >
> > > Note that all previous feed-forward GS architectures struggle with scalability. Mamba’s limited state space struggles to remember long sequences, transformers face quadratic time complexity, and cost volume methods risk memory issues at high resolutions. Even our hybrid 7M1T design fails at our highest resolution without additional strategies.
> > >
> > > Long-LRM tackles this large-scale input processing with efficient token merging and Gaussian pruning (Sec. 3.3 and 3.4). By addressing the significant redundancy from overlapping Gaussians caused by high resolution and long sequences, it further reduces memory usage by removing unnecessary Gaussians. Its novel pruning strategy combined with opacity regularization maintains gradient flow during training, achieving substantial memory savings without compromising performance.
> > >
> > > Additionally, training on large-scale is inherently slow and resource-intensive (section 4.2). Training with our largest-scale setting is 4 times slower than our smallest scale (table 3). To avoid wasting GPU resources, we devised an effective curriculum training schedule (section 4.2) to facilitate training on large-scale datasets with high-resolution inputs.
> > >
> > > Finally, our goal is to deliver a **simple yet effective solution** for reconstructing high-resolution, wide-coverage scenes with GS. While the final design appears straightforward, this simplicity is deliberate and the result of **extensive exploration**. We opted for our final model configuration for its **simplicity**, and abandoned several additional constructs we tried  that could have given marginal improvements yet with greater complexity. We believe our approach provides a simple and scalable solution with strong performance, which constitutes a meaningful technical contribution.

---

> > > > ### Comment · Reviewer_K9N5 · 2024-11-28
> > > > **Reply to authors**
> > > >
> > > > Thanks for  your responce. I need to clarify the two points:
> > > >
> > > > 1. What I mean about real time is the response delay for the new input. Your methods needs to process 32 input images all together with 1.3s. When the new input images coming in, this creates a minimum latency of 1.3s between when the new image arrives and when you get its result. This is not real time.
> > > >
> > > > 2. Feed-forward approaches in both NeRF and GS were originally developed to address sparse-view settings, where optimization-based methods struggle with limited input data. In your dense-view scenarios with 16/32 views, optimization-based methods like Scaffold-GS already achieve superior reconstruction quality compared to the proposed feed-forward approach. While the authros argue that the contributions are orthogonal to optimization-based methods, it's better to demonstrate that combining both approaches leads to new SOTA results or at least help optimization-based approaches.

---

> ### Author Response · Authors · 2024-11-22
>
> 7. Whether Long-LRM can seamlessly concatenate predicted Gaussians from multiple sets of images.
>
> **Ans**: Currently, Long-LRM is not trained to seamlessly fuse Gaussians predicted from multiple sets of images. Optimization-based methods like CityGaussian achieve this by first optimizing a coarse set of Gaussians for the entire scene, which serves as a prior. The scene is then divided into smaller blocks and optimized independently, which can be directly concatenated thanks to the reference provided by the coarse Gaussians.
>
> In contrast, Long-LRM, as a feed-forward method, predicts Gaussians from a set of images in a single pass without any additional knowledge of concurrently or previously generated Gaussians from other sets. Seamlessly concatenating all sets of Gaussians may require additional post-processing steps, such as filtering and merging techniques. An exciting future direction would be to further scale feed-forward GS models to handle hundreds of input images directly.

---

> ### Author Response · Authors · 2024-12-02
>
> Long-LRM is not designed for an online reconstruction setting. It does not anticipate users sending images one by one to acquire reconstructions on a per-frame basis. Our focus has always been on the offline setting, where a user provides all input images at once, and the model outputs a global reconstruction, same as the optimization-based methods. We emphasize that our method, with the same experiment setting, achieves a 700x speed-up compared to all optimization-based methods, delivering competitive quality with only a 0.8dB drop compared to Scaffold-GS, while outperforming the original 3D GS and Mip-Splatting. While the method is not online, we argue that, in the context of offline setting, this is a substantial improvement over a 10-min wait required by the optimization-based techniques.
>
> In addition, we respectfully disagree with the reviewer's claim that rendering quality should be the sole standard for publishing an academic paper. In the field of 3D vision and scene reconstruction, speed and efficiency are widely recognized as critical contributions. Many highly impactful works, such as SNeRG (ICCV 2021) and MobileNeRF (CVPR 2023), were published with a primary focus on reconstruction or rendering efficiency improvements, despite not achieving state-of-the-art rendering quality. Both papers were highly impactful and widely recognized for their efficiency contributions, with SNeRG accepted as an Oral in ICCV 2021 and MobileNeRF accepted as an award candidate in CVPR 2023.
>
> Similarly, our work demonstrates a clear advantage in reconstruction efficiency, achieving a significant speed-up of two orders of magnitude with minimal performance loss relative to state-of-the-art methods. We hope the reviewer will follow the community standards in evaluating our work and recognize the importance of improvements in efficiency. Unbiased reviewing should focus on overall relative advancements rather than requiring perfection across all metrics for a research paper.
>
> Furthermore, as the reviewer noted, feed-forward approaches were traditionally only demonstrated in sparse-view settings. Our work is the first feed-forward model to show competitive quality to optimization-based methods in a dense-view wide-reconstruction scenario. This extends the scope of what feed-forward methods can achieve. We view this as an important first step and look forward to future works that build on this direction to achieve even better quality, potentially reaching state-of-the-art rendering quality.
>
> Finally, we would like to mention that for Scaffold-GS, its specialized GS representation stores features and offsets instead of spherical harmonics coefficients, making direct initialization with Long-LRM's predicted Gaussians incompatible, while initializing with only point positions would lead to suboptimal results. Therefore, we consider Scaffold-GS's contribution orthogonal to our study that is based on the original 3D GS representation. Exploring advanced feed forward models to directly predict Scaffold-GS-based features and offset parameters is an interesting direction for future research.

---

### Official Review · Reviewer_JaZB · 2024-10-28

**Soundness:** 3
**Presentation:** 3
**Contribution:** 2
**Rating:** 6
**Confidence:** 5

**Summary:**

This paper proposed a generalizable 3D reconstruction framework for a long range of input images. 3D Gaussian Splatting is used as the 3D representation like many previous works. The network architecture is design as mixture of Mamba2 and transformer layers to process input tokens with long length.  The whole model is trained in single stage and can reconstruct large scale 3D secenes on DL3DV-140, Tanks and Template datasets.

**Strengths:**

1. Extends the application of feed-forward 3D scene reconstruction to longer-range inputs.
2. Sound network architecture design by combining transformers and Mamba2 to process long token sequences.
3. Applies a token merging module to reduce computational overhead for processing long-range input views.
4. The author provides justification for using Mamba in Table 2, although the comparison with GS-LRM is somewhat unfair.

**Weaknesses:**

1. Insufficient justification for using Mamba. GS-LRM claims it can accept arbitrary input view numbers by downsampling images with large patch sizes to shorten the overall token length for global attention. The features after attention can then be upsampled to predict a large number of Gaussians. However, in Table 2, the authors provide the same patch size for both the 7M1T and GS-LRM architectures, leading to an unfair comparison.

2. Why not cost volumes and abadon 3D inductive biases. Although the authors have argued that methods like MVSplat are prone to out-of-memory (OOM) issues, these challenges are largely engineering problems that can be addressed with techniques like FlashAttention or through lightweight network architecture design. The authors need to clarify this point; otherwise, this work may mislead the feed-forward 3D scene reconstruction community.

3. Lack of discussion on 3D reconstruction works utilizing Mamba. This work seems to overlook prior research utilizing Mamba for 3D reconstruction, such as Hamba, Gamba, and MVGamba, which have been publicly available for over six months. Discussing these related works is necessary to emphasize the motivation for this study.

4. Limited technical contribution and insight. The technical contributions of this work are quite limited, as are its insights. It mainly extends previous 3D reconstruction efforts that use Mamba for scene reconstruction. Notably, combining Mamba and transformer blocks cannot be considered novel, as this setup was proposed in the Mamba v2 paper and has been widely adopted in various Vision Mamba works with Mamba v2. When evaluating the technical contributions, it is challenging to provide a positive rating, as nearly all modules in this paper have been widely used in numerous feed-forward 3D object reconstruction studies over the past year. Additionally, the claims regarding cost volumes contradict established practices in feed-forward 3D scene reconstruction.

Based on the weaknesses outlined above, I think this paper is marginally below the acceptance threshold.

**Questions:**

please refer to the weakness part.

---

> ### Author Response · Authors · 2024-11-22
>
> 1. Justification for using Mamba.
>
>
> **Ans**: We do not find any claim in the GS-LRM paper about "accepting arbitrary input view numbers by downsampling images with large patch sizes to shorten the overall token length for global attention." After confirming with the GS-LRM authors, we believe they do not make this claim. Specifically, GS-LRM paper only shows results with no more than six input views. Our experiments in Tab. 2 also demonstrate that scaling GS-LRM to handle our large input scale is impractical.
>
> Note that GS-LRM adopts a similar multi-resolution training strategy, from 256x256 low-resolution to 512x512 high-resolution images, and uses the same patch size of 8 across these resolutions. For our comparisons in Table 2, we adopted the same resolution and patch settings as GS-LRM for the two stages of training in the 2nd and 3rd blocks. We believe this ensures a fair comparison by reusing most of the hyperparameters selected by GS-LRM. With GS-LRM’s same patch size, the use of Mamba is critical for scaling up to longer input sequences.
>
> Additionally, we found that smaller patch sizes generally yield better reconstruction quality. For example, increasing the patch size to 16 for GS-LRM with four input views (the first setting in Table 2) results in a PSNR of 20.85, lower than the original PSNR of  21.13 with a patch size of 8. We retrain using GS-LRM’s original patch size of 8, ensuring both fair comparisons and high-quality results.
>
>
> 2. Justification for abandoning 3D inductive biases like cost volumes.
>
>
> **Ans**: GS-LRM has demonstrated the effectiveness of transformer-based models without relying on cost volumes or other 3D inductive biases. Building on this foundation, our work aims to scale LRM models to support long input sequences, which we believe is a promising and important direction for addressing the problem of large scene-level 3D reconstruction.
>
> We also found that global attention methods, including both GS-LRM and Long-LRM, can significantly outperform cost volume-based methods. To illustrate this, we present a quality comparison on the RealEstate10k dataset with the standard two input views in the table below. This dataset is widely used by prior works, and the baseline results are taken directly from their respective papers. We retrained two versions of Long-LRM (w/ and w/o token merging) on the same training dataset for this comparison. The results show that global attention methods, both GS-LRM and Long-LRM, achieve significantly better performance compared to other baselines with more 3D inductive biases, including the cost volume-based MVSplat and the epipolar line-based pixelSplat.
>
> **Rendering quality on RealEstate10k with 2-view input**
> | Method | PSNR(↑) | SSIM(↑) | LPIPS(↓) |
> | -------- | :-------: | :-------: | :-------: |
> | pixelSplat | 25.89 | 0.858 | 0.142 |
> | MVSplat | 26.39 | 0.869 | 0.128 |
> | Long-LRM (w/ token merge) | 27.26 | 0.872 | 0.130 |
> | GS-LRM | 28.10 | 0.892 | 0.114 |
> | Long-LRM (w/o token merge) | 28.44 | 0.893 | 0.113 |
>
> Additionally, we argue that cost volumes are not inherently memory-efficient for long sequences and high resolutions. Naively, one needs to construct a 3D high-resolution cost volume for every input image, requiring aggregating features from all other co-visible images, and then fuse the volumes to produce consistent Gaussians for the entire scene. This approach is computationally intensive and memory-demanding for large-scale scenes. Notably, previous feed-forward GS methods employing cost volumes (e.g., MVSplat, MVSGaussian) have only demonstrated results at 256 resolution with 1–4 input views. Successfully scaling up these models to handle long-sequence inputs remains a challenging research problem rather than a straightforward engineering task.
>
> Overall, the task of feed-forward reconstruction for wide-coverage large scenes is still a new and underexplored area. Our work provides the first solution to this challenge, but we do not claim that global attention or Mamba-based models are the only possible solutions. It is impractical to explore every alternative in a single submission. We look forward to future explorations in this direction, including advanced cost volume-based methods and lightweight network designs, to tackle this challenge.

---

> ### Author Response · Authors · 2024-11-22
>
> 3. Discussion on prior Mamba-based 3D reconstruction methods, such as Hamba, Gamba, and MVGamba.
>
>
> **Ans**: We thank the reviewer for pointing out the missing citations. We will include and discuss these relevant works—Hamba, Gamba, and MVGamba—in a revision. In particular, Hamba is a Mamba-based method specifically designed for hand reconstruction, while Gamba and MVGamba utilize Mamba for feed-forward 3D GS modeling, which aligns more closely with our task. However, these methods adopt pure Mamba architectures and focus on object-level sparse-view reconstruction, typically using only 1–4 input images and processing around 10K patch tokens. In contrast, Long-LRM is designed for scene-level reconstruction from a significantly larger number of input images (up to 32) with higher resolutions, presenting key new challenges in processing long-sequence input tokens. Instead of relying on a pure Mamba architecture, Long-LRM employs a hybrid architecture that combines transformer and Mamba2 blocks, enabling efficient processing of **up to 250K tokens**. This scale is far beyond what previous Mamba-based methods handle. Our contributions specifically target the scalability and the unique requirements of large scene-level reconstructions with long-sequence inputs, which go beyond the scope of methods like Gamba, MVGamba, and Hamba.
>
>
>
> 4. Technical contribution.
>
>
> **Ans**: We emphasize that our work targets a **new and under-explored task**: achieving **feed-forward, wide-coverage 3D scene reconstruction from a large number of input images**. Prior to our work, large scene-level NeRF or 3DGS reconstructions could only be achieved by per-scene optimization methods. Recent feed-forward methods, such as GS-LRM, Gamba, and MVGamba, have made progress but focus on object-level reconstruction or sparse input views, leaving wide-coverage scene reconstruction largely unexplored. Our work is the first to demonstrate that feed-forward models can be effectively designed to handle large-scale scene reconstruction tasks, which we believe provides valuable insights to the community.
>
> A key aspect of our contribution is addressing the challenge of **long-sequence inputs** with up to ~250K tokens—a scale more than ten times larger than existing transformer- or Mamba-based 3DGS models. This significantly longer sequence introduces unique challenges in maintaining high reconstruction quality while managing computational efficiency.
>
> To tackle these challenges, we propose a novel **hybrid architecture** that combines transformers and Mamba2 blocks, achieving a balance between quality and efficiency. While similar hybrid architectures have been applied to other vision tasks, **we are the first to adopt this hybrid approach for 3D reconstruction**. In addition, we introduce **token merging** and **Gaussian pruning techniques** to further reduce computational complexity, enabling our method to handle significantly longer sequences compared to prior methods.
>
> We also emphasize that training at this large input scale is inherently more challenging and unstable than training on shorter sequences. To address this, we designed a multi-resolution training strategy with combined loss functions, enabling stable training at large scales. To the best of our knowledge, **this is the first feed-forward 3D reconstruction solution demonstrated at this large input scale with up to 32 images at a resolution of 960x540**. Furthermore, we conducted a comprehensive study and analysis of various design choices at the same large input scale; we hope this provides helpful insights into the trade-offs and performance of different techniques for long-sequence reconstruction tasks.
>
> We hope our work inspires the community to develop new technologies to address the challenges of the under-explored task of feed-forward wide-coverage 3D scene reconstruction.

---

> > ### Comment · Reviewer_JaZB · 2024-11-28
> >
> > I appreciate authors detailed rebuttal. Experiments presented here have addressed my main concerns about this work so i raise my rating as 6.

---

### Official Review · Reviewer_1Xmy · 2024-11-03

**Soundness:** 2
**Presentation:** 3
**Contribution:** 2
**Rating:** 5
**Confidence:** 5

**Summary:**

This paper introduces Long-LRM, a feed-forward model for large-scale 3D Gaussian reconstruction that can process 32 high-resolution (960×540) input images in just 1.3 seconds on a single A100 GPU. The key innovation lies in its hybrid architecture combining Mamba2 and transformer blocks, along with efficient token merging and Gaussian pruning strategies to handle long sequences and memory constraints. The model achieves 600× faster reconstruction than optimization-based 3D Gaussian Splatting while maintaining comparable or better quality, as demonstrated through comprehensive evaluations on DL3DV and Tanks and Temples datasets.

**Strengths:**

1. Enhance feed-forward scene reconstruction methods, eg, GS-LRM to more input views.

2. The usage of hyrbid network of Mamba and transformer is reasonable for handling extreme long-sequence tokens, though it is not the first paper in this field that introduce Mamba.

3. Practical solutions for memory efficiency through token merging and Gaussian pruning, enabling scaling to high resolutions (960x540) where other variants fail.

4. The ablation study is comprehensive, well demonstrating the effectiveness of each component, with clear metrics on performance gains.

**Weaknesses:**

1. Lack of novelty. The core contribution of this paper seems a combination of GS-LRM and Hamba, Gamba  and MVGamba.

2. The lack of discussion on the above Mamba-based 3D reconstruction models, which have been publicly available more than half years, is not acceptable.

3. While this paper presents several practical innovations in memory optimization, it may be more suitable for computer vision conferences rather than ICLR.

**Questions:**

While the work on token merging and Gaussian pruning for memory efficiency is valuable, these engineering optimizations better fit computer vision venues like CVPR. Despite the technical advances,  could you justify how this work aligns with ICLR's focus on fundamental machine learning methodology rather than computer vision conferences?

---

> ### Author Response · Authors · 2024-11-22
>
> 1. Technical novelty.
>
> **Ans**: We emphasize that our work targets a **new and under-explored task**: achieving **feed-forward, wide-coverage 3D scene reconstruction from a large number of input images**. Prior to our work, large scene-level NeRF or 3DGS reconstructions could only be achieved by per-scene optimization methods. Recent feed-forward methods, such as GS-LRM, Gamba, and MVGamba, have made progress but focus on object-level reconstruction or sparse input views, leaving wide-coverage scene reconstruction largely unexplored. Our work is the first to demonstrate that feed-forward models can be effectively designed to handle large-scale scene reconstruction tasks, which we believe provides valuable insights to the community.
>
> A key aspect of our contribution is addressing the challenge of **long-sequence inputs** with up to ~250K tokens—a scale more than ten times larger than existing transformer- or Mamba-based 3DGS models. This significantly longer sequence introduces unique challenges in maintaining high reconstruction quality while managing computational efficiency.
>
> To tackle these challenges, we propose a novel **hybrid architecture** that combines transformers and Mamba2 blocks, achieving a balance between quality and efficiency. While similar hybrid architectures have been applied to other vision tasks, **we are the first to adopt this hybrid approach for 3D reconstruction**. In addition, we introduce **token merging** and **Gaussian pruning techniques** to further reduce computational complexity, enabling our method to handle significantly longer sequences compared to prior methods.
>
> We also emphasize that training at this large input scale is inherently more challenging and unstable than training on shorter sequences. To address this, we designed a multi-resolution training strategy with combined loss functions, enabling stable training at large scales. To the best of our knowledge, **this is the first feed-forward 3D reconstruction solution demonstrated at this large input scale with up to 32 images at a resolution of 960x540**. Furthermore, we conducted a comprehensive study and analysis of various design choices at the same large input scale; we hope this provides helpful insights into the trade-offs and performance of different techniques for long-sequence reconstruction tasks.
>
> We hope our work inspires the community to develop new technologies to address the challenges of the under-explored task of feed-forward wide-coverage 3D scene reconstruction.
>
> 2. Discussion on publicly available Mamba-based 3D reconstruction models.
>
> **Ans**: We thank the reviewer for pointing out the missing citations. We will include and discuss these relevant works—Hamba, Gamba, and MVGamba—in a revision. In particular, Hamba is a Mamba-based method specifically designed for hand reconstruction, while Gamba and MVGamba utilize Mamba for feed-forward 3D GS modeling, which aligns more closely with our task. However, these methods adopt pure Mamba architectures and focus on object-level sparse-view reconstruction, typically using only 1–4 input images and processing around 10K patch tokens. In contrast, Long-LRM is designed for scene-level reconstruction from a significantly larger number of input images (up to 32) with higher resolutions, presenting key new challenges in processing long-sequence input tokens. Instead of relying on a pure Mamba architecture, Long-LRM employs a hybrid architecture that combines transformer and Mamba2 blocks, enabling efficient processing of **up to 250K tokens**. This scale is far beyond what previous Mamba-based methods handle. Our contributions specifically target the scalability and the unique requirements of large scene-level reconstructions with long-sequence inputs, which go beyond the scope of methods like Gamba, MVGamba, and Hamba.

---

> ### Author Response · Authors · 2024-11-22
>
> 3. Submission to a computer vision conference vs ICLR.
>
> **Ans**: ICLR has consistently served as a great venue for works at the intersection of machine learning and its applications, including computer vision. Our work falls into this category, advancing machine learning methods for 3D computer vision. Notably, several foundational papers on 3D large reconstruction models, such as LRM, Instant3D, PF-LRM, and DMV3D, have been published at ICLR, with three of them recognized as Oral or Spotlight papers (full paper titles listed below). They all address similar tasks in 3D reconstruction and generation, demonstrating the strong relevance of this area to the ICLR community. As a follow-up to the LRM series of ICLR papers, our Long-LRM tackles the new challenge of processing long-sequence inputs for wide-coverage scene reconstruction using novel machine learning techniques, which is essential for scaling feed-forward 3D reconstruction to large scenes. We believe this aligns well with the scope of ICLR and hope our work will benefit the community.
> * LRM: Large Reconstruction Model for Single Image to 3D, ICLR 2023 (Oral)
> * Instant3D: Fast Text-to-3D with Sparse-view Generation and Large Reconstruction Model, ICLR 2023
> * PF-LRM: Pose-Free Large Reconstruction Model for Joint Pose and Shape Prediction, ICLR 2023 (Spotlight)
> * DMV3D: Denoising Multi-view Diffusion Using 3D Large Reconstruction Model, ICLR 2023 (Spotlight)

---

> > ### Comment · Reviewer_1Xmy · 2024-11-25
> >
> > Thank you for the detailed rebuttal and major revision. I have several points for further discussion.
> >
> > Regarding the venue selection between CV conferences and ICLR: First, I would like to clarify that the LRM papers you referenced was published by Adobe Research at ICLR 2024, not ICLR 2023. The significance of that work lies in its pioneering contribution to the 3D community, demonstrating that a deterministic reconstruction model can achieve convergence and generalization on large-scale 3D datasets. Over the past year, the 3D community has produced approximately 20 LRM-related papers. From the current perspective, I believe adapting GS-LRM to broader inputs and scenarios may not align with ICLR's scope. **However, I suggest leaving this decision to the AC and other reviewers' collective judgment.**
> >
> > My second point concerns the exploitation of Gaussian's explicit properties. Following the approaches of LGM and GS-LRM, it's feasible to use different Gaussians for different scene views and merge them afterward. This would eliminate the need to process an excessive number of Gaussians simultaneously, which aligns with real-world large scene rendering where scenes are typically divided into blocks based on camera parameters.
> >
> > I am raising my score from 3 to 5, as the authors have adequately addressed my concerns regarding the discussion of related work. I look forward to hearing other reviewers' perspectives before making a final decision.

---

> > > ### Author Response · Authors · 2024-12-04
> > >
> > > We believe our work is clearly aligned with ICLR's scope. Given the strong disagreement, we will leave the judgment about our work's relevance to the Area Chairs and other reviewers, as noted by the reviewer.
> > >
> > > We respectfully disagree with the claim that directly merging Gaussians from sparse-view methods could achieve the same effect as Long-LRM. For example, in GS-LRM, global attention across all input images is crucial before Gaussian merging, ensuring information exchange among views. This process cannot be replicated by independently predicting Gaussians for each view and simply concatenating them (see the example below). Independent predictions lack the view-dependent information critical for harmonized global reconstruction. As seen in multiple visualizations, 3D GS methods fail to deliver accurate rendering when test views deviate significantly from input views. Long-LRM’s high-capacity processing is essential for handling dense, high-resolution inputs and achieving coherent scene reconstruction.
> > >
> > > To illustrate our point, we adopted a GS-LRM model trained with 4 input views on DL3DV10K, and performed 4-view local reconstruction of a test scene in DL3DV140. Images of the local Gaussians are shown here: https://huggingface.co/longgggglrm/llrm_stuff/resolve/main/misc/sparse.pdf?download=true
> > >
> > > From the images, we can see that independent local GS reconstructions from local subsets of views exhibit significant noise at the boundaries and in unseen regions. Simply concatenating these reconstructions results in severe floaters and noticeable discontinuity artifacts.

---

### Author Response · Authors · 2024-11-24
**Revised pdf uploaded**

We have completed the revisions to our paper based on the reviewers' suggestions and uploaded the updated version. Major revisions (highlighted in red) include:

* included citation and discussion of Gamba, MVGamba and Hamba
* updated experiment numbers from Long-LRM trained on the full DL3DV-10K dataset, instead of only the first 7K
* added comparison results with previous feed-forward GS methods on RealEstate10K
* added comparison results with 3D GS variants
* expanded the ablation study with additional results on the training objectives

We thank all reviewers for their feedback, which helped us improve the paper.

---

### Meta-Review · Area_Chair_eoqL · 2024-12-21

**Metareview:**

This paper proposes a feed-forward model for large-scale 3D Gaussian reconstruction that can process 32 high-resolution (960×540) input images in 1.3 seconds on a single A100 GPU. The key innovation lies in its hybrid architecture combining Mamba2 and transformer blocks, along with efficient token merging and Gaussian pruning strategies to handle long sequences and memory constraints.  Reducing memory consumption through token merging and Gaussian pruning should be appreciated.  Also, 600x faster reconstruction performance than optimization-based 3D Gaussian Splatting is significant.  On the other hand, the reviewers raised concerns regarding limited novelty, insufficient discussion to justify the proposed model, and insufficient validation.  Reviewer 1Xmy was concerned with whether the presented work is aligned with the scope of ICLR.  In the rebuttal, the authors have provided additional experiments to address the concerns on validation and argued the justification for using Mamaba and differences from prior Mamba-based 3D reconstruction methods.  Possibility of directly merging Gaussians and positioning of the paper in relation to Scaffold-GS are further discussed between the authors and the reviewers.  In the end, two reviewers are negative while one reviewer is positive for the paper.  Regarding the concern on the venue selection, AC acknowledges ICLR is appropriate.  Since the size of the community is rapidly growing and the scope of ICLR is expanding, not restricted to fundamental machine learning methodology, this paper should not be rejected because of out-of-scope.  On the other hand, although the authors claim that this work targets scalability of large scene-level reconstruction with long-sequence, arguments on scalability seem not sufficient.  There should be argument that 32 input images are enough to be long-sequence.  In fact, Reviewer K9N5 raised concern on the possibility of real large-scale scene reconstruction.  Through the author-reviewer discussion, it turned out that handling even more input images (say, hundreds) is not possible and needs further work.  At least in-depth discussion toward real large-scale reconstruction should be addressed. Otherwise, the paper cannot be scientifically solid in the context. Providing comparison with Mamba-based 3D reconstruction methods (which may need some post-processing to adjust the same scale) will make the paper stronger.  Reviewer K9N5 also suggested to combine 3D GS variants such as Scaffold-GS into the proposed method, which seems to achieve new SOTA; however, the authors just excuse that the contribution of Scaffold-GS is orthogonal and complementary.  The usage of hybrid network of Mamba and transformer has brought significant speed-up with on-par quality, but offline setting is not attractive for the applicability of the method in real scenarios. AC finds room to strengthen the method to be solid for real large-scale scene reconstruction.  On balance, the remaining concerns outweigh the current technical contributions and the paper would benefit from more work.  For this, the paper cannot be accepted to this conference.

**Additional Comments On Reviewer Discussion:**

See above.

---

### Decision · Program_Chairs · 2025-01-22

Reject